# A Review of Atrial Fibrillation Detection Methods as a Service

**DOI:** 10.3390/ijerph17093093

**Published:** 2020-04-29

**Authors:** Oliver Faust, Edward J. Ciaccio, U. Rajendra Acharya

**Affiliations:** 1Department of Engineering and Mathematics, Sheffield Hallam University, Sheffield S1 1WB, UK; 2Department of Medicine—Cardiology, Columbia University, New York, NY 10027, USA; ciaccio@columbia.edu; 3Ngee Ann Polytechnic, Electronic & Computer Engineering, Singapore 599489, Singapore; aru@np.edu.sg; 4Department of Bioinformatics and Medical Engineering, Asia University, Taichung 41354, Taiwan

**Keywords:** atrial fibrillation, computer-aided diagnosis, service design

## Abstract

Atrial Fibrillation (AF) is a common heart arrhythmia that often goes undetected, and even if it is detected, managing the condition may be challenging. In this paper, we review how the RR interval and Electrocardiogram (ECG) signals, incorporated into a monitoring system, can be useful to track AF events. Were such an automated system to be implemented, it could be used to help manage AF and thereby reduce patient morbidity and mortality. The main impetus behind the idea of developing a service is that a greater data volume analyzed can lead to better patient outcomes. Based on the literature review, which we present herein, we introduce the methods that can be used to detect AF efficiently and automatically via the RR interval and ECG signals. A cardiovascular disease monitoring service that incorporates one or multiple of these detection methods could extend event observation to all times, and could therefore become useful to establish any AF occurrence. The development of an automated and efficient method that monitors AF in real time would likely become a key component for meeting public health goals regarding the reduction of fatalities caused by the disease. Yet, at present, significant technological and regulatory obstacles remain, which prevent the development of any proposed system. Establishment of the scientific foundation for monitoring is important to provide effective service to patients and healthcare professionals.

## 1. Introduction

Atrial Fibrillation (AF) is the most common supraventricular arrhythmia, which affects about 1% of the population [1,2]. There are 4.5 million confirmed cases in the EU and 2.3 million in the U.S. Over 5% of persons aged 65 years or older have AF [3,4]. Other statistics show that approximately 3.7–4.2% of persons aged between 60 and 70 years and 10–17% of those aged 80 years or older suffer from this arrhythmia [2,5]. This prevalence increase continues into old age where over 17% of people aged 85 or above are AF patients [6]. It is estimated that by 2050, AF may affect 17.9 million people in the EU and 6.12 million in the U.S. [7,8]. This increase is partly due to demographic trends, such as population aging, but it also reflects better survival rates of persons with predisposing conditions to AF [9]. To prevent or at least dampen this increase, it is recommended to screen for AF in high-risk patients regardless of age and to screen all patients aged 65+ years during primary care visits [10,11].

AF is characterized by a rapid, irregular heartbeat [12]. Common causes are: hypertension, cardiomyopathy, ischemia, and rheumatic heart disease [13]. AF can be a chronic condition that is an independent risk factor for heart failure, myocardial infarction, dementia, cognitive dysfunction, and ischemic stroke, leading to increased morbidity and mortality [14,15,16,17,18]. The irregular beating may cause blood clots, which increases stroke risk five-fold [19,20] and risk of death two-fold, when compared with individuals of the same age, independent of other risk factors [21,22]. There are on average 40 AF-related strokes every day in England [23]. The symptoms and complications lead to a substantial burden of disease [14,24,25,26]. Inpatient AF care accounts for more than two-thirds of the annual direct costs of AF [27,28,29]. Anticoagulation is an effective therapy for managing AF patients who are at risk of stroke and can reduce its risk [30,31]. Studies, based on hospital data, found that approximately 30% of stroke patients known to have AF prior to admission were taking anticoagulants [32,33]. Unfortunately, the pathophysiological mechanisms of AF are not completely understood, which makes the diagnosis difficult [34]. The problem is even more pressing for patients suffering from paroxysmal AF, because there may be no symptoms at early onset, and there is spontaneous termination of arrhythmia [35]. An estimated 20% of all AF cases may go undetected since the disease is asymptomatic [36,37,38]. Under such circumstances, treatment is not initiated, which is likely to result in negative outcomes for patients [39,40]. Therefore, the development of a new AF detection technology could be assistive in screening and treatment, reducing morbidity and mortality [41]. This could help to address the large percentage of time and money spent by health services throughout the world [42].

There has recently been a recognizable change in healthcare, the introduction of the Internet of Things (IoT), Smart Devices, and Big Data, which are altering the way people live [43]. This digitalization promises to improve both efficacy and efficiency in existing healthcare processes through continuous measurements and real-time data analysis [44]. Small and power efficient sensing hardware and analysis algorithms in software drive these innovations [45]. The ongoing establishment of mobile healthcare devices was recognized by the World Health Organization (WHO) [46]. The organization recognizes the potential benefits of m-health devices to enable patient-led data acquisition, networked care delivery, real-time treatment, and adherence monitoring [47,48]. Patient-led data acquisition means the measurement setup can be established with minimal training. That requires non-intimidating patient-facing technology that is simple to use and at the back end of compliance management. In their review, Giebel and Gissel established that m-health devices can be used for AF detection [49]. Having the technological framework to communicate and process physiological data creates the need for disease-specific analysis algorithms to deliver a service that can improve outcomes for patients [50]. AF detection is an important application area for m-health devices, because of the substantial medical need and the relative ease with which the measurement data can be acquired [51,52,53,54].

In this review, we promote the idea that the AF detection problem can be solved with service design. From a medical perspective, we need to extend the observation window in order to increase the AF detection rate. A sensible way of doing this is to integrate the data acquisition process into a patient’s day-to-day activities. Such patient-led data acquisition mandates smart systems that are able to cope with the normal living environment. Designing such smart systems requires an outline of how a user, in this case the patient, interacts with the physical problem solution. Once established, the outline is refined into a service that offers the required functionality. The service architecture takes the form of interconnected e- and m-health systems, which establish data communication, storage, and processing. This architecture allows us to reason about the problem on a cognitive rather than an engineering level. To be specific, the problem becomes one of making sense of the acquired data and implementing decision processes that create the AF detection service.

To establish AF detection as a service, we review data processing and decision-making paradigms. The review starts with background information on Electrocardiogram (ECG) and RR interval signals that can be used to detect AF. With that foundation, we can frame the problem according to how the patient interacts with the service, as a timeline diagram. The timeline helps to establish the information flow, which is shaped by executing signal processing and decision-making algorithms. Section 3 introduces these algorithms. The ability to decide if a particular signal shows signs of AF is crucial for service performance. Therefore, we review computer-aided AF diagnostic systems in the Discussion section. The Conclusion summarizes the review methods and states the key results of our investigation.

## 2. Background

In this section, we discuss the different measurements that can be used to support AF diagnosis. From a service perspective, patient-led data acquisition is an important requirement. Imaging methods, such as echocardiogram [55,56,57,58] and chest X-ray [59,60], are unsuitable for measurements in the patient environment. Blood test biomarkers can be useful for risk prediction [61]. However, it is not yet feasible to carry out either measurement or analysis in the home environment.

ECG and RR interval signals result from passive non-invasive measurements. These measurements can be carried out in the patient environment. ECG results from measurement of the electric field emitted by heart [62]. The resulting signals provide a good base for automated AF detection [63,64,65]. This is because AF symptoms, such as poorly coordinated atrial activation and turbulent cardiac beating [66], alter the signal morphology [67,68,69]. RR interval signals can be extracted from the ECG, and these signals are useful for detecting heart rhythm disorders, such as AF [70]. The following sections provide further details concerning both ECG- and RR interval-based AF detection. Wearable Photoplethysmogram (PPG) might provide an alternative for measuring the RR intervals [71].

### 2.1. Photoplethysmogram

PPG is an optical measurement that detects variations in light reflected from human tissue [72]. The main feature of the resulting signal is the peripheral pulse, which is synchronized with the R peaks in the ECG. Once the PPG signal is captured, the beat-to-beat interval is measured and saved as an RR interval signal [73]. Hence, the AF detection methods are similar to those that process RR intervals extracted from ECG. The measurement can be done with mobile devices, such as wrist watches [74]. As such, the PPG measurement becomes an add-on feature for a mobile device, which keeps the cost down. Widespread availability, combined with little extra cost for the sensor, might lead to a significant increase in data volume. Given the right detection algorithms, that data can help to establish the presence of AF in the wider population.

For user comfort, these devices can move around, which is likely to cause activity specific artifacts in the measurements [75]. These artifacts mandate preprocessing algorithms that clean up the signal for the AF detector. In 2017, Sološenko et al. addressed the lack of annotated public PPG databases by generating a photoplethysmogram model from the well-known MIT–BIHAF database [76]. This model attempts to reflect the cardiovascular system; hence, there might be variation in how well that model fits for a particular patient. That uncertainty also affects the AF detection methods that were developed with the photoplethysmogram model. Furthermore, detectors with a high computational complexity might exploit some of the hidden assumptions that went into the photoplethysmogram model design. In a practical scenario, these assumptions might not hold, which reduces the detector performance. Another approach is to record PPG alongside ECG and transfer the labels, established by a cardiologist, from ECG to PPG [71,77]. Such double recording is necessary, because PPG signals cannot be used for diagnosis. In other words, cardiologists diagnose AF based on ECG signals. Until that changes, PPG-based AF detection can only be used in gadgets that can raise the suspicion that AF symptoms are present. The medical diagnosis will require additional measurements.

### 2.2. Electrocardiogram

Twelve-lead ECG is the reference standard for AF detection [78]. In order to establish a diagnosis, the signal is analyzed by a trained clinician [34]. Figure 1 shows two ECG signal segments from PhysioNet’s AF database [79,80]. A normal ECG cycle is referred to as Normal Sinus Rhythm (NSR). It consists of a P wave, QRScomplex, and T wave. The first plot in Figure 1 provides an example of NSR. In that plot, the first P wave is marked and labeled. Pathologically, AF can be described as a breakdown of organized atrial electrical activity [30]. This breakdown manifests itself in the absence of a P wave in the ECG signal and the R peak appearing at irregular intervals [81]. The second plot in Row 1 of Figure 1 provides an example of an ECG signal that shows signs of AF. The R peaks appear at irregular time intervals, and the P wave is muted or absent. Table 1 provides a summary of studies that investigated computer support for automated AF detection in ECG signals.

The morphology of normal ECG signals differs from person-to-person [82]. ECG-based AF diagnosis means to detect disease-related morphology changes. The main challenge is that these morphology changes may not be unique, which can lead to reduced specificity in identifying the disease that causes these changes. Extending the observation time might help to solve the sensitivity problem [83,84]. Furthermore, the relationship between atrial activity, measured by surface ECGs, and AF mechanisms, is not yet well understood [85]. This ambiguity translates into inaccurate detection algorithms [86]. Detecting AF based on the symptomatic heart rhythm change could be a helpful way forward.

### 2.3. Heart Rate

Heart Rate (HR) can be used to detect AF episodes [87]. The second row in Figure 1 shows the RR intervals that correspond to the ECG signals shown in the first row. As such, RR interval signals are constructed from RR intervals. These RR intervals represent the time from one R peak in the ECG signal to the next. In Figure 1, this is indicated by the arrows from the ECG to the RR interval signal.

For a human expert, it might be difficult to detect the subtle linear and nonlinear changes in the RR interval trace that indicate AF. Therefore, manual analysis of RR interval measurements may result in inter- and intra-observer variability. Digital biomarkers and algorithmic decision support can help to avoid these problems and hence improve diagnostic quality [88,89].

The occurrence of symptomatic episodes in paroxysmal AF is uncertain [90,91]. Some episodes may last more than 48 h [90]. Statistical analysis shows that during AF, RR intervals have a larger standard deviation and a shorter correlation length than those during NSR. Hence, these measures can be used as digital biomarkers for AF detection. However, they are not specific enough when it comes to differentiating AF from other arrhythmias [92,93].

## 3. Materials

The materials were arranged in accordance with the service design methodology [94]. Our approach was human-centered; we detail the experience we wanted to create for the people who are affected by the AF detection service. Through a collaborative effort, we refined this user experience into a sequence of interrelated actions, which were visualized and orchestrated through a timeline. The individual actions were accomplished by signal processing algorithms, and these algorithms are reviewed in Section 3.2.1 and Section 3.2.2.

### 3.1. Service

There is a clearly identified need from healthcare trusts for automated detection and management of AF patients [42,95]. The main objective for AF detection as a service is to increase the observation duration. For this review, we propose a service architecture that accomplishes that through patient-led data acquisition. The patient lives with an ECG or RR interval sensor, and the measurement results are constantly analyzed for signs of AF. Interventions happen only when AF is diagnosed.

To progress with the service design, we translated this story into a timeline that structured the interrelated actions that made up the service. Figure 2 shows the timeline, and it indicates the actions for cardiologist, patient, and AF detection service. The timeline starts with a nurse signing up a patient for the AF detection service, and the patient-led data acquisition process begins. The measurement data are communicated, stored, and processed by cloud technology [96]. This establishes the AF monitoring functionality. The cardiologist is informed if the processing system detects AF symptoms in the data. The cardiologist can review the available data and fuse this information with knowledge from personal interactions with the patient to reach a diagnosis [97]. In case the cardiologist rejects the findings from the AF detection unit, the monitoring continues. If AF is diagnosed, the patient is informed, and treatment can begin. The AF service then monitors the treatment.

### 3.2. System Architecture

With the appropriate IoT infrastructure in place, we can focus on the cognitive aspects of establishing AF detection as a service. These cognitive aspects are concerned with refining the information that flows from the patient to the cardiologist. The diagram, shown in Figure 3, depicts the patient as the information source and the cardiologist as the information sink. There are two distinct routes for the information. The first route indicates information extraction through Deep Learning (DL). The second route incorporates information extraction through digital biomarker algorithms. The extracted information can serve as a quantifiable indicator for the cardiologist during the diagnosis process. Apart from this direct use, the extracted information can also serve as input for classical Machine Learning (ML). The ML algorithms refine that input into a single score. In the next sections, we review the digital biomarker, DL, and classical ML algorithms used to achieve the information refinement.

#### 3.2.1. Digital Biomarkers

The concept of the digital biomarker is an important foundation component in this review. Digital biomarkers represent some form of information that was extracted from an underlying measurement [98,99]. Algorithms are used to extract the information [98]. In the past, digital biomarkers were seen as complementing traditional methods, such as human interpretation of ECG [100]. Since it is possible to establish the digital biomarker quality through statistical methods and machine classification, they have started to gain acceptance [101].

Digital biomarkers for AF detection may be grouped into nonlinear, time, frequency, and time frequency. Figure 4 provides a taxonomy of the individual groups and the following list provides more details on the methods used to extract the digital biomarkers:Time-domain biomarkers result from algorithms that analyze signals over time. They can be used to model the dynamic behavior of the human heart. These are the most explainable digital biomarkers, because they quantify some of the observations that could be gained through visual inspection of the signal. However, these biomarkers fail to explain the nonlinear characteristics of heart’s oscillation process. Hence, they do not provide all the available information. Examples of time-domain biomarker algorithms are differential equations [102], Principal Component Analysis (PCA) [103,104], the Hadamard transform [105,106], Linear Discriminant Analysis (LDA) [107], and Independent Component Analysis (ICA) [108].Heart rhythm is associated with the frequency content of the signal. Frequency-domain digital biomarkers extract measures that indicate the characteristics of the rhythm change. They are especially sensitive to rhythm change caused by AF. Essentially, frequency analysis methods are linear. Hence, frequency-domain digital biomarkers fail to reflect nonlinear properties that might be a relevant source of information. Algorithms that extract these biomarkers are based on the theory of Fourier Transform (FT) and Power Spectral Density (PSD) [109].Mixed-domain digital biomarkers are based on the idea of combining the analytical strength of multiple domains [110]. Most practical realizations are based on the fact that the spectrum changes as time progresses [111]. It is possible to track spectral changes over time [112]. Based on the premise that a rhythm disorder will emerge in the spectral domain, time resolution allows us to estimate the time of an AF event [113]. These biomarkers are a great tool for explaining the arrhythmia locations. However, the analysis does not go beyond linear relationships. Examples for mixed domain transformation algorithms are: Discrete Wavelet Transform (DWT) [110], Discrete Cosine Transform (DCT) [114], Continuous Wavelet Transform (CWT) [115], Wavelet Packet Decomposition (WPD) [116], Stationary Wavelet Transform (SWT), Short-Time Fourier Transform (STFT) [117,118], Empirical Mode Decomposition (EMD) [119], and Time-Varying Coherence Function (TVCF) [120].Nonlinear digital biomarkers aim to reflect the nonlinear nature of the human heart [121,122]. Surrogate data tests [123] show that nonlinearity cannot be ruled out in ECG and RR interval signals. Performance testing of nonlinear digital biomarkers, with statistical and classification methods, indicates that they are an independent information source. Examples of algorithms that provide parameters for such biomarkers are: Recurrence Quantification Analysis (RQA) [124,125], Higher Order Spectra (HOS) [126], Fractal Dimension (FD) [127], entropy [128], energy [129], and Largest Lyapunov Exponent (LLE) [130].RQA defines a set of digital biomarkers that result from analyzing recurrence plots. Quantifying both the number and duration of recurrences in the physiological signal can help us to understand the phase space trajectory. Changes in the phase space trajectory might indicate the presence of AF [124]. The HOS is a spectral representation of third-order and higher moments and cumulants. It yields nonlinear digital biomarkers that quantify the nonlinear correlations between several frequency components [131,132,133,134]. These measures are sensitive to rhythm changes, which make them useful for arrhythmia detection. FD and LLE are chaos measures of signal complexity [12,132]. Entropy gauges the information content of a signal [135,136,137,138]. In general, signals with less structure, such as RR interval signals, have more information as compared with signals that have more structure, like the ECG. Energy biomarkers quantify the rhythmicity and regularity of signals.

#### 3.2.2. Artificial Intelligence

Artificial Intelligence (AI) algorithms can assist in establishing the input signal class. There are three main types of classification algorithms: (a) supervised, (b) unsupervised, and (c) reinforcement [139]. Each algorithm type requires tailored and specific design steps. Figure 5 shows a taxonomy of AI systems management. Unsupervised learning establishes the data structure through clustering. A new data vector is classified by establishing the cluster to which it most likely belongs. Examples of classification algorithms based on unsupervised learning are: k-means clustering [140] and self-organizing maps [141]. Reinforcement learning algorithms train continuously by taking into account feedback on the quality of the decision. Examples for such reinforced learning algorithms are hidden Markov models for decision-making [142] and generative adversarial networks [143].

Supervised learning follows the idea of a student-teacher relationship. Labeled data can be used to train and test supervised learning algorithms. The standard method to test the decision reliability is 10-fold cross-validation [144,145]. Holdout, also known as blindfold cross-validation, establishes the classifier performance under practical conditions [146]. Examples of classical ML algorithms based on supervised learning are: Naive Bayes (NB) [144], Probabilistic Neural Network (PNN) [147], Support Vector Machine (SVM) [148,149,150,151], Random Forest (RF) [152], Levenberg–Marquardt Neural Network (LMNN) [153,154], K-Nearest Neighbor (K-NN) [155], Decision Tree (DT) [156], rule-based [157], and RandomForest (RF) [158].

DL is based on the concept that a complex structure is required to represent a complex scenario. A Deep Neural Network (DNN) mimics the human brain in terms of functionality and complexity. In the past, algorithms, such as the Artificial Neural Network (ANN), modeled the functionality of a small number of interconnected neurons [136]. These systems have the ability to learn by updating weight parameters within the individual neurons. However, ANN lacked complexity, and therefore, these algorithms cannot readily handle high-dimensional data. In contrast, DL has the complexity of extracting knowledge from high-dimensional data [159]. For example, it is possible to input RR interval data directly into a DL system. ANN algorithms require dimension reduction through digital biomarkers before they can process the data. DL was successfully applied to a wide range of applications, including image analysis [160,161,162], sound classification [163], and signal analysis [164]. DNNs are primarily composed of three layer types: convolutional, pooling, and fully-connected. A convolutional layer is an adaptive filer, the weights of which are updated during training. The pooling layer performs an outcome-directed dimension reduction of the values that flow through the network. Max pooling is a common approach [165]. The fully-connected layers are usually at the end of the deep neural network processing chain. They concentrate the data, which are flowing through the network, such that individual labels emerge [166]. There exists a wide range of named DNN variants, such as Recurrent Neural Network (RNN) [167], Convolutional Neural Network (CNN) [168], Deep Belief Network (DBN) [169], and Long Short-Term Memory (LSTM) [170]. In Figure 5, we refer to ANN and other algorithms described in the previous paragraph as classical ML. These algorithms, as well as the DNN algorithms described in this paragraph belong to the set of supervised learning algorithms. The difference between classical ML and DL is that the former set of algorithms require digital biomarkers for dimension reduction, and the latter can process physiological signals directly.

## 4. Discussion

Improving AF detection rates is a technological problem that arises from the fact that in some cases, detectable AF symptoms are transient. The problem can be solved in two ways: (1) improve the sensitivity of the detection method [171]; (2) extend the observation time [34]. Improving the AF detection sensitivity is a gradual process that is based on analyzing and understanding the physiological data [172,173,174]. The role of technology to speed up that process is limited. In contrast, technology is the main driver when it comes to extending the observation time. The interplay between understanding of the disease and the availability of technology resulted in four main approaches when it comes to extending the observation duration.

Holter monitors have been fitted with more memory and better energy measurement to extend the measurement time. State-of-the-art devices achieve 72 h ECG recordings in clinical practice. Data analysis and indeed a diagnosis happens after the measurement when the device is returned to the stroke clinic. This process usually incurs a delay of two to three weeks before the diagnosis is communicated to the patient.Patient-triggered event recorders are a way to circumvent the energy and storage limitations of Holter monitors. These devices depend on a patient suspecting an AF episode and acting on this suspicion by triggering a measurement. Therefore, such patient-triggered event recorders have a high intra- and inter-observer variability. During a state of reduced consciousness, e.g., sleep, the patient cannot trigger the recorder. The consistency with which the device is used varies from patient to patient, dependent on their mental capability [175]. For example, elderly patients might not be able to interact with the devices to initiate the required measurements [176]. The ideas of patient-triggered event recorders underpin a number of commercial products, such as CardiaMobile from AliveCor (web page (12.04.2020): https://www.alivecor.com/kardiamobile) and Apple Watch (web page (12.04.2020): https://support.apple.com/en-us/HT208955). The services these devices offer are aimed at the patient, and as such, they can only establish a suspicion that AF is present. A diagnosis requires further measurement, in most cases with Holter monitors.Cardiac event recorders have replaced patients with algorithm-based triggers, thereby addressing the same energy and storage limitations while reducing intra- and inter-observer variability. The idea is machine algorithms never sleep, and they arrive at the same conclusion for the same input. The diagnosis support quality of these devices is directly related to the sensitivity and specificity of the algorithms used to trigger the recording. These performance measures tend to go up when the computational complexity of the algorithm is increased. Unfortunately, computational complexity is also positively correlated with the energy consumption of the processing platform that executes the algorithm. Hence, these cannot overcome the energy limitation completely. The trade-off between energy consumption and computational complexity of the algorithm has to be resolved during design time. Another limitation of this recording device is the delay between the time when an event was recorded and the diagnosis. During clinical studies, patients wore event triggers, such as the ER910AF Cardiac Event Monitor from Braemar (web page (12.04.2020): http://braemarllc.com/event_products.html), for 30 days [177]. That prolonged observation duration increased the AF detection five-fold.Mobile cardiovascular telemetry employs state-of-the-art IoT technology to communicate the measurement signals to a central cloud server [178,179]. In that cloud server, the data are stored and processed with an AI system. The results and indeed the measurement data are made available for the reading cardiologist. Figure 6 provides an overview diagram that details the interplay of technologies used for AF detection. Storing the data on a central location is beneficial when it comes to retraining the AI model. One possible way to facilitate the retraining is to use the diagnosis from a cardiologist to label specific segments of the data on the cloud server. This will increase the available knowledge, which can be used to retrain AI models in order to make them more robust.

Having the technology to measure, communicate, store, and process the physiological signals enables us to focus on AF detection as a service. Figure 2 shows how the service provides value for both the patient and reading cardiologist. To be specific, the ability to detect AF in the measurement data can help during the initial diagnosis and for treatment monitoring. The service quality is directly related to the performance of the algorithms used to detect signs of AF in the measurement data.

Table 1 summarizes studies published on AF detection in ECG signals. The AF detection performance is reported in terms of Accuracy (Acc), Sensitivity (Sen), and Specificity (Spe). All these studies followed the information flow diagram shown in Figure 3. Patient data came in the form of digitized ECG stored in databases. MIT-BIH and CinCchallenge databases are all publicly available on PhysioNet [79,80], whereas “measurement data” indicates a proprietary database. All studies, apart from Yuan et al. 2016 [180], employed digital biomarkers to reduce data dimensionality such that they could be classified with classical ML algorithms. This process is also referred to as extracting relevant information from the signal. Unfortunately, during the design process, it was difficult to differentiate between relevant and irrelevant information [164]. Hence, selecting a set of biomarkers would result in information loss, and the classical ML algorithm was not presented with all of the available information. Therefore, classical ML algorithms performed well during design time when the biomarker selection reflected the available data. However, these algorithms underperformed when they were presented with more and more varied data.

Table 2 summarizes studies published on AF detection in RR interval signals. In the majority of cases, that work was based on the same publicly available databases as the studies on ECG, because RR intervals can be extracted from ECG; see Section 2.3. The two most recent studies from Ivanovic et al. 2019 [181] and Faust et al. 2018 [182] were based on DL, and all other studies depended on classical ML techniques to detect AF in the RR interval signals.

Table 1 and Table 2 indicate that the performance results, achieved by the classification algorithms, for ECG and RR intervals, were similar. For example, both Wang et al, 2020 [183], and Faust et al., 2018 [182], used the same MIT-BIH AFDBdatabase, and they achieved 98.8% and 98.51% accuracy, respectively. Having similar performance results on the same database is significant, because it indicates that it is possible to look beyond classification performance when choosing between ECG and RR intervals. From the technology perspective, the fact that RR interval signals have a significantly lower data rate when compared to ECG make them the signals of choice for the AF detection service. To give an indication of the data rate difference between ECG and RR intervals, we used the signals shown in Figure 1 as a reference example. The amount of data required for each of the four second ECG traces was:(1)$$\mathrm{ECG}\;\mathrm{data}=w\times f_{s}\times r=5\;s\times 250\;\mathrm{Hz}\times 12\;\mathrm{bit}=15000\;\mathrm{bit}$$
where *w* is the window length, $f_{s}$ is the sampling frequency, and *r* is the resolution of the signal converter used for the measurement. In contrast, the number of bits for the RR interval plots was:(2)$$\begin{array}{cc} \mathrm{HR}\;{\mathrm{data}}_{\mathrm{NSR}} & =nrr\times d=4\times 9\;\mathrm{bit}=36\;\mathrm{bit}\\ \mathrm{HR}\;{\mathrm{data}}_{\mathrm{AF}} & =8\times 9\;\mathrm{bit}=72\;\mathrm{bit} \end{array}$$
where nrr is the number of RR intervals in the observation window and *d* is the number of bits used to encode the RR interval. This example shows that the ECG signal required at least 200 times more data. This difference translated into the fact that RR interval communication, storage, and processing are significantly less expensive than ECG. Furthermore, RR interval sensors tend to be smaller than ECG sensors, because they have a lower hardware complexity and require less energy.

Potential cost saving is only one aspect for choosing between ECG and RR intervals. AF detection as a service will be used by medical professionals. As it stands, ECG is the reference standard when it comes to AF diagnosis. Using RR intervals for that task constitutes a significant paradigm shift. Furthermore, human interpretation of RR interval signals is difficult. Even digital biomarkers, extracted from RR interval signals, become difficult to track over time, because their value might depend on the AF progression state [213]. Hence, an RR interval-based AF detection service does not meet all the requirements from the reading cardiologists. A combination of ECG and RR interval signals might lead to a feasible solution, where RR intervals are used for real-time analysis and ECG is only captured when there is a suspected AF event. Such a system would combine the cost efficiency of RR interval processing with the human readability of ECG.

### Open Questions and Future Work

Data are central when it comes to designing and verifying AF detection methods as a service. As such, the publicly accessible databases are an invaluable tool during the creation of AF detection methods, because those data can be used to design, validate, and compare different detection algorithms. Unfortunately, both the amount and diversity of publicly accessible data are insufficient to represent measurements from all possible patients. Hence, it is important that future studies on AF detection methods use all the available data (MIT-BIH NSR/AFDB/LTAFDB, CinC challenge 2017, and THEW). In general, the current databases and indeed the current algorithms are just a starting point when it comes to AF detection as a service. To determine a way forward, we have to look at areas where AI is applied successfully. Most notably is speech recognition, which is employed in digital assistance systems. These systems implement dynamic deep learning systems, i.e., classification algorithms that constantly improve through human supervision. This will help to diversify the machine knowledge to include an awareness of different types of AF and symptoms of AF being superimposed on signals showing signs of other diseases. Digital biomarkers will be instrumental with regard to human supervision, because they extract specific information from the signal that might assist in validating a deep learning result.

Despite recent progress, described in the previous sections, barriers for heart health monitoring as a service still remain. These barriers originate from the fields of technology, financial, regulatory, cultural, and behavioral change. Upfront investments for heart health monitoring as a service are considerable, despite the fact that the technology is infrastructure available. The cost drivers are technology integration, ensuring safe and reliable data exchange between the IoT components. Furthermore, regulation dictates what technology can be used. This might lead to the deployment of inefficient or outdated technologies. From a patient perspective, the main consideration is health, which will depend on the ability of the digital processing system. The fast pace of digital development benefits innovation, but at the same time imposes barriers in terms of the high level of hardware turnover. Older devices might not support the latest service functionalities, and this might be caused by business rather than technological considerations.

As outlined in the previous paragraph, regulation is a major barrier when addressing the institutional implementation of heart health monitoring services. With respect to new technology, many healthcare trusts have adopted a fluent position in their approach to policy and regulation. With respect to data, the culture is changing. There is a greater acceptance to provide data for specific benefits. In terms of heart health monitoring as a service, the data are physiological measurements, and the benefits are health assessments. However, fundamental concerns still remain about the data security, safety, and reliability of the service. Hence, it requires time for a service to be accepted. With respect to heart health monitoring as a service, individual behavioral changes should gradually cause cultural change. Mobile technology, such as the smartphone, will become central to health and well-being.

Innovative e-health and m-health technologies, such as atrial fibrillation detection as a service, may raise ethical concerns that need to be addressed in order to provide consistent regulations and guidance to healthcare professionals and developers [44]. Trust and concern about confidentiality and security are major issues to consider when regarding patient-led data acquisition [214]. This is particularly relevant for wireless and Internet-based data transmission, because the data can be intercepted and then used to harm the patient. For example, auto upload of data must only be done with patient consent and in a format that lacks patient identifiers. Hence, there are also psychological barriers that prevent the uptake even if the advantages outweigh the drawbacks [215]. A factor that plays a role in technology adoption is the capability of the user. From a technical perspective, a high data quality must be ensured in order to build trust in the technology [216]. Furthermore, system complexity must be reduced as much as possible in order to ensure barrier-free access to the technology and indeed the benefits it brings.

In the future, the infrastructure that facilitates AF detection as a service can be extended to detect other arrhythmias as well. State-of-the-art DL models can help provide diagnostic support for multiple arrhythmias based on ECG signals [55,217,218,219].

## 5. Conclusions

We started this review by establishing the need for a human-centered AF detection as a service. Through service design exploration, we distilled the available knowledge about the disease and the available treatment methods into the problem of increasing observation time. By proposing patient-led data acquisition coupled with IoT technology and hybrid decision-making, we proposed a sustainable concept to solve that problem. The smartness of the service originates from the processing algorithms involved in the information refinement. We established the current state of research by reviewing digital biomarker extraction and medical decision support algorithms. These algorithms take center stage when it comes to cardiologist support for AF detection. By reviewing the performance of these decision support systems, we established the feasibility of the service concept for AF detection.

In this review, we focused on RR interval- and ECG-based AF detection. We established that both signals could be used to detect AF symptoms. Hence, both signals can provide service as a basis for diagnosis. A human-centered and sustainable AF detection service mandates that the signal data travel from the patient to a central location for processing, as well as diagnostic support. RR interval signals are easier to measure, process, communicate, and store as compared to ECG data because they have a significantly lower data rate. Thus, an RR interval-based AF detection service will be more cost effective and likely more convenient for the patient. Therefore, monitoring RR interval signals with an AF detection service can provide a viable solution for the problem of increasing the AF detection rate. Evidence suggests that a higher detection rate will lead to more treatment. This will have a positive impact on patient health and reduce stroke risk.

## Figures and Tables

**Figure 1 ijerph-17-03093-f001:**
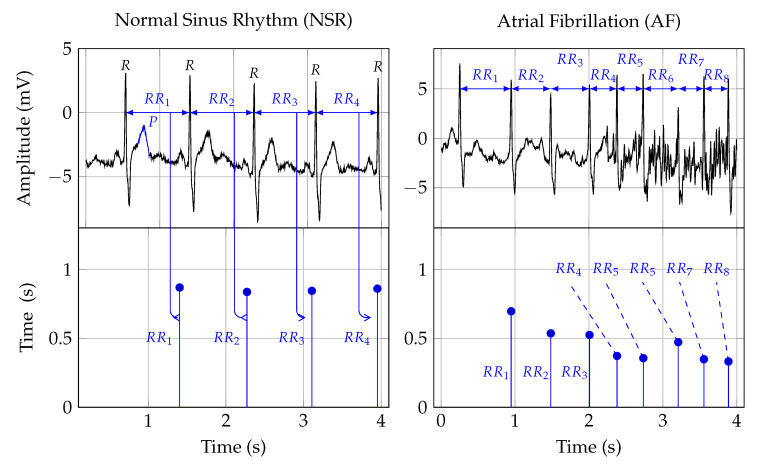
The first row depicts two ECG signals and the second row the corresponding RR interval traces. The two plots in the first column show NSR, and the plots in the second row show AF symptoms. The *R*peak, in the ECG plots, indicates ventricular depolarization, i.e., the time of the heart beat. The time between two heart beats, say *a* and *b*, is indicated as the *RR*interval. That time duration forms the amplitude, and the time location of the second beat *b* is the time location of an RR interval sample. The *P*wave, labeled in the NSR ECG plot, indicates atrial depolarization.

**Figure 2 ijerph-17-03093-f002:**
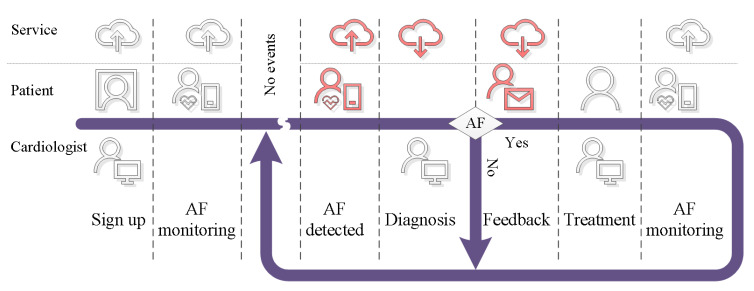
Service timeline.

**Figure 3 ijerph-17-03093-f003:**
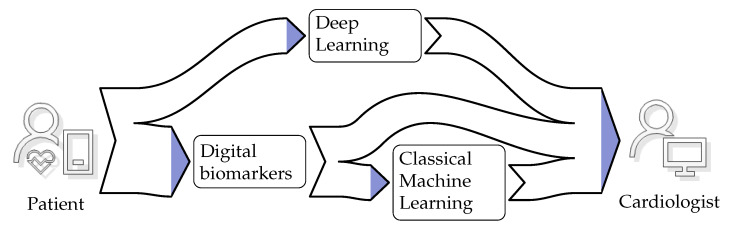
Information flow from patient to cardiologist.

**Figure 4 ijerph-17-03093-f004:**
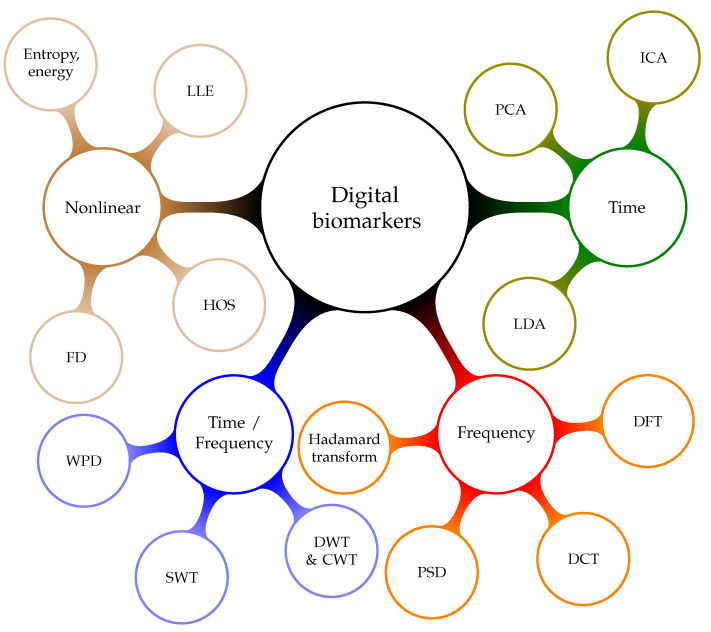
Taxonomy of digital biomarkers.

**Figure 5 ijerph-17-03093-f005:**
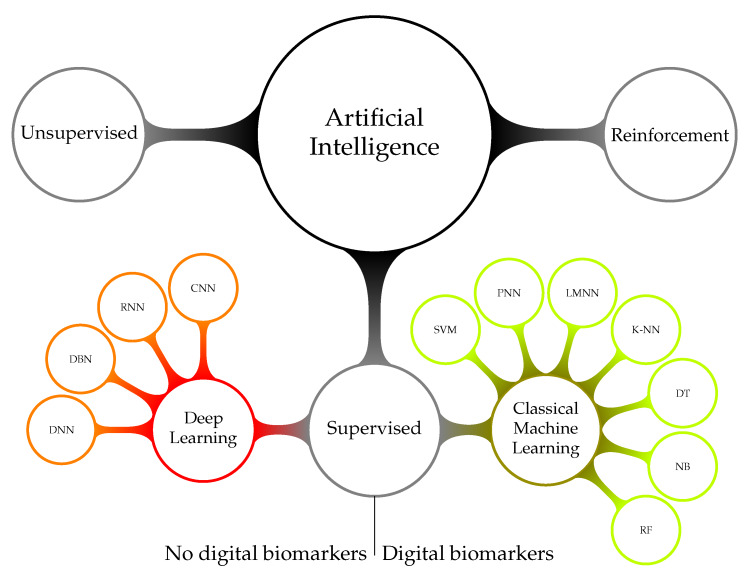
Taxonomy of artificial intelligence.

**Figure 6 ijerph-17-03093-f006:**
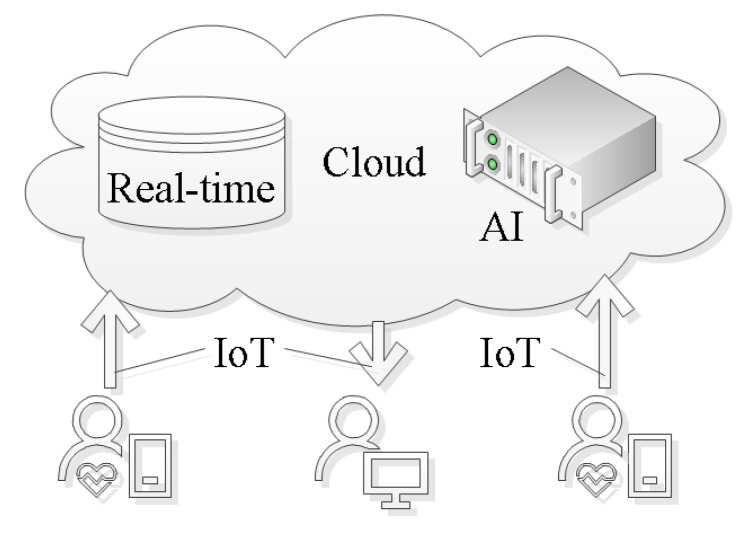
Technologies that underpin atrial fibrillation detection as a service.

**Table 1 ijerph-17-03093-t001:** A summary of automated AF detection in ECG signals. A “+” in the Salient features column indicates a positive point. Conversely, a “−” indicates a negative point.

Authors	Data	Digital Biomarkers	AI	Performance in %			Salient Features
				Acc	Sen	Spe	
Wang et al., 2020 [183]	MIT-BIHAFDB	WPD followed by multivariate statistical features	ANN	98.8	98.7	98.9	+ Good model performance + Cross-validation used − Digital biomarkers required − Only one database used − No blind fold validation
Cao et al., 2020 [184]	CinCchallenge 2017	Data augmentation	DNN	78.35	-	-	+ Improving small datasets − Real measurement data are required − Classification models might exploit weaknesses in the augmentation
Marsili et al., 2019 [185]	MIT-BIH AFDB and measurements	Shannon entropy	Threshold	98.1	99.2	97.3	+ Hardware implementation + Fast and energy efficient − Local decision making − Impossible to verify the decision
Yao et al., 2019 [186]	CinC challenge 2017	DWT	Multi-scale CNN	98.18	98.22	98.11	+ Good model performance − High computational complexity − One type of digital biomarker
Lui et al., 2018 [187]	MIT-BIH AFDB/NSR/Arrhythmia Database	Normalized fuzzy entropy	Threshold	-	-	-	+ Focused study − One type of digital biomarker − No AI
Xia et al., 2018 [188]	MIT-BIH AFIB	STFT, SWT	CNN	98.63	98.79	97.87	+ Focused study − One type of digital biomarker − No AI − No validation
Kora et al., 2017 [189]	MIT-BIH AFIB	CS-SCHT	LMNN	99.30	96.97	99.43	+ Good model performance − Classical machine learning − No validation
Tripathy et al., 2017 [138]	MIT-BIH AFIB	Sample entropy, VMD	DBN	98.27	98.80	97.77	+ Good model performance + Index to combine multiple digital biomarkers − Classical machine learning − No validation − Only one database used
Annavarapu and Padmavathi, 2016 [190]	MIT-BIH Arrhythmia Database	CS-SCHT	LMNN	99.50	99.97	98.70	+ Good model performance − Classical machine learning − No validation
Abdul-Kadir et al., 2016 [102]	MIT-BIH NSR/AFDB	Dynamic system	ANN, SVM	95.00			+ Cross-validation used − Linear digital biomarkers − Classical machine learning
Yuan et al., 2016 [180]	MIT-BIH NSR/AFDB/LTAFDB	-	Autoencoder DL	98.31	96.56	99.04	+ Good model performance + Deep learning − No validation
Asgari et al., 2015 [135]	MIT-BIH AFDB	SWT, Log-energy entropy, peak-to-average power ratio	SVM	97.10	97.00	97.10	+ 2-fold cross-validation − Classical machine learning
Daqrouq et al., 2014 [191]	MIT-BIH AFIB	WPD	PNN	97.92	-	-	+ 2-fold cross-validation − Classical machine learning − No outcome directed digital biomarker selection
Martis et al., 2013 [192]	MIT-BIH AFDB/Arrhythmia Database	DWT	NB	99.33	99.32	99.33	+ 10-fold cross-validation + Multiple arrhythmias + Noise considerations − Classical machine learning
Majia et al., 2013 [193]	MIT-BIH Arrhythmia Database	HOS and EMD	Thresholded	-	96	-	+ Classic approach − Single decision border − Only one database − No validation
Rincón et al., 2012 [194]	MIT-BIH AFIB	Statistical measures	Fuzzy classifier	-	98.09	91.66	+ P-wave detection − Classical machine learning − Only one database − No validation
Lee et al., 2012 [41]	MIT-BIH NSR/AFDB	RMSSD, sample entropy, Shannon entropy	Threshold	98.44	97.63	99.61	+ M-health + Event recorder − Local decision making
Fukunami et al., 1991 [195]	Measurement data	Frequency-domain	Statistical analysis	-	91	76	+ Classic manuscript − No benchmark test − No AI
Parvaresh and Ayatollahi, 2011 [196]	MIT-BIH AFIB	Autoregressive model	Statistical classifier	-	96.14	93.20	+ Early paper − Machine learning − Linear digital biomarkers

**Table 2 ijerph-17-03093-t002:** A summary of automated AF detection in RR interval signals.

Authors	Data	Digital Biomarkers	AI	Performance in %			Salient Features
				Acc	Sen	Spe	
Ivanovic et al., 2019 [181]	Measurement data	-	DL	89.67	94.20	-	+ Three class problem + Cross-validation used + Detailed model description − No signal detrending − No benchmark data
Anderson et al., 2019 [197]	MIT-BIH NSR/AFDB/Arrhythmia Database	-	CNN and RNN	-	98.98%	96.95%	+ Three databases + Cross-validation used − No signal detrending
Faust et al., 2018 [182]	MIT-BIH AFDB	-	LSTM	98.51	98.32	98.67	+ First deep learning detector based on RR intervals − No signal detrending − One database
Henzel et al., 2017 [198]	MIT-BIH AFDB	Linear measures	Threshold	93	90	95	+ Insight into digital biomarkers − No AI − One database
Cui et al., 2017 [199]	MIT-BIH NSR/AFDB	Ensemble model	Threshold	97.78	97.04	97.96	+ Insight into digital biomarkers + Two databases − No AI
Islam et al., 2016 [200]	MIT-BIH AFDB	Entropy	Threshold	96.38	96.39	96.38	+ Insight into digital biomarkers − No AI − One database
García et al., 2016 [201]	MIT-BIH AFDB/Arrhythmia Database	RWE, SWT	Threshold	93.32	91.21	94.53	+ Insight into digital biomarkers + Two databases − No AI
Kennedy et al., 2016 [202]	SPIT^1^, THEW^2^, MIT-BIH Arrhythmia Database/AFDB/ LTAFDB/NSR/SVA	Sample entropy, CV, RMSSD, MAD	RF	-	92.80	98.30	+ Most data + Cross-validation − Traditional machine learning was used to gain insight into digital biomarkers
Petrėnas et al., 2015 [203]	MIT-BIH NSR/AFDB	Linear measures	Threshold	-	97.1	98.3	+ Insight into digital biomarkers + Two databases − No AI
Andersson et al., 2014 [204]	MIT-BIH Arrhythmia Database/AFDB	Statistical measures	Threshold	-	94.6	95.8	+ Insight into digital biomarkers + Two databases − No AI
Zhou et al., 2014 [205]	MIT-BIH Arrhythmia Database/AFDB/NSR	Shannon entropy	Threshold	96.05	96.72	95.07	+ Insight into digital biomarkers + Two databases − No AI
Lee et al., 2013 [136]	MIT-BIH Arrhythmia Database/NSR	TVCF, Shannon entropy	Threshold	-	98.20	97.70	+ Insight into digital biomarkers + Two databases − No AI
Lake and Moorman 2011 [69]	MIT-BIH AFDB	Sample entropy	Threshold	-	91	94	+ Insight into digital biomarkers − No AI − One database
Lian et al., 2011 [68]	MIT-BIH Arrhythmia Database/AFDB/ NSR/NSR2	Statistical measures	Threshold	-	95.9	-	+ Insight into digital biomarkers + Three databases − No AI
Lake and Moorman 2011 [69]	MIT-BIH AFDB	Sample entropy	Threshold	-	91	94	+ Insight into digital biomarkers − No AI − One database
Mohebbi et al., 2011 [206]	MIT-BIH AFDB	RQA, recurrence plot	SVM	-	97.00	100.00	+ Insight into digital biomarkers − One database
Yaghouby et al., 2010 [67]	MIT-BIH Arrhythmia Database	Statistical and geometrical	Genetic algorithm	99.11	-	-	+ Insight into digital biomarkers − One database
Huang et al., 2010 [86]	MIT-BIH AFDB/NSR	Linear	Threshold	-	96.10	98.10	+ Insight into digital biomarkers + Two databases − No AI
Dash et al., 2009 [63]	MIT-BIH AFDB	TPR, RMSSD, Shannon entropy	Threshold	-	90.20 - 94.40	91.20 - 95.10	+ Insight into digital biomarkers − No AI − One database
Babaeizadeh et al., 2009 [64]	MIT-BIH AFDB	Statistical measures	DT	-	98	-	+ Insight into digital biomarkers − One database
Ghodrati and Marinello, 2008 [207]	MIT-BIH Arrhythmia Database	Probability density function	Threshold	-	92	-	+ Insight into digital biomarkers − No AI − One database
Ghodrati et al., 2008 [208]	Measurement data, MIT-BIH Arrhythmia Database/AFDB	Normalized absolute deviation and normalized absolute difference	Threshold	-	89	-	+ Insight into digital biomarkers + Two databases − No AI
Kikillus et al., 2007 [209]	MIT-BIH NSR/AFDB	Linear, Frequency and RQA	Threshold	-	94.1	93.4	+ Insight into digital biomarkers + Two databases − No AI
Logan and Healey, 2005 [210]	MIT-BIH AFDB	Statistical measures	Threshold	-	94.4	97.2	+ Insight into digital biomarkers − No AI − One database
Tateno and Glass, 2001 [211]	MIT-BIH AFDB	CV & Kolmogorov–Smirnov test	Threshold	-	91.20	96.08	+ Insight into digital biomarkers − No AI − One database
Artis et al., 1991 [212]	MIT-BIH AFDB	Statistical measures	ANN	-	92.86	-	+ Insight into digital biomarkers − No AI − One database − No validation

^1^ No further information on the database provided in the paper, ^2^ (web page (13.04.2020): http://thew-project.org/).

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
