# Peer review of "A Review of Atrial Fibrillation Detection Methods as a Service"

_ijerph, 2020, doi:10.3390/ijerph17093093_

Round 1
Reviewer 1 Report
Paper is a review of current work related to arythmia detection. Authors concentrate on ANN in recent forms (DL, ML etc. ) approach to be considered as the main algorithm that supports arythmia detection. The major problem in such approach is still patient. Any proposed system that relies on patient discipline and self-control might give unpredictealbe results.
The system that allows more optimal feedback about any self - control over the patient is in general leading to increase diagnostic system performance.
Also Authors understand current limitations in such system application due to regulatory changes. However regulatory changes must follow system proposal. If sciencists develop tool that is unknown, and proof its reliablity and performance this might be a spark to make regulatory changes.
I'm impressed with literature, that Authors followed in order to create paper.
Reviewer 2 Report
This manuscript offers a comprehensive review of atrial fibrillation detection methods based on ECG signals and RR intervals, upon which the authors promote the idea of a AF system as a service. The motivation is strong, and makes for a very interesting paper: the idea to promote a human-centred AF monitoring system makes sense considering the reviewed literature and is, overall, a very laudable effort. Despite this, the manuscript seems a little superficial, especially on the presentation of the state-of-the-art methods, so it needs a deeper and more nuanced discussion on the past, current, and future research in AF detection, the current weaknesses of the methods and how to overcome them.
Comments:
(1) This is a very good and extensive review of the literature. Tables 1 and 2 are especially valuable. Although section 3.2.1 and 3.2.2 offer an overview of possible methods for the task at hand, there is no discussion from the authors regarding what would be the best method(s) for AF detection. It would be interesting to see a deeper discussion into the most promising methods, including their strengths and weaknesses, and ideas on how to improve them.
(2) I would like to see more discussion on the potential of less expensive ECG acquisition systems for the human-centred part of the system? There are currently alternatives to medical ECG acquisition, (like the bitalino hardware or some ECG/PPG systems now embedded in certain smartwatches) that could be used for easy and frequent signal acquisition, at home by the patients. How would this affect the authors’ idea of AF as a patient-centred service? Would the much greater availability of data affirm the PPG as a good alternative to ECG?
(3) I believe the manuscript would benefit very much from a section on the currently available databases for research in AF. Some are mentioned in Tables 1 and 2, but no details are given. What databases would the authors recommend for a prospective researcher? Are the databases adequate or should researchers work towards more data in more adequate settings?
(4) It would be interesting to know in more detail what is the authors vision on the future of AF detection. Specifically, what should be the focus of research in the near future? What are the issues that need to be tackled immediately? Should we focus on deep learning or digital biomarkers, when and why? What are the weaknesses that are common to most state-of-the-art methods, and how can we mitigate them?
Reviewer 3 Report
This paper presents a review on the methods that can be used to efficiently and automatically detect AF in RR interval and ECG signals. The paper is well organized and the review is well addressed. However, it is suggested to add a subsection to describe the ECG databases considered in the methods surveyed.
Reviewer 4 Report
Dear Authors,
I would first congratulate you for your extensive review about the techniques for automatic AF detection. I have just a few suggestions, listed below:
- Line 18: I would add “sustained” after “common”
- Figure 1: Please, explain in the figure legend, what the abbreviation “NSR” comes from.
- Please, explain the abbreviations in the legends of all other figures. It would be more convenient for the readers.
Round 2
Reviewer 2 Report
I thank the authors for reading and answering my comments. However, the changes to the manuscript have been very minor. I believe that the manuscript is still very superficial and could do with more deep expert insights from the authors regarding the various aspects of AF recognition. Beyond the idea of AF detection as a patient-centred service, the review should not be just a summary of literature information, but a window to the authors’ knowledgeable perspectives on the current landscape and future research opportunities.
1 - I disagree with the author’s perspective that discussing potential improvements/solutions/research paths limits the scope of the review. Actually, if done right, it would make it less superficial and more valuable.
2 - The discussion on PPG and the extraction of RR intervals from PPG presented in the cover letter is interesting, but unfortunately, it was not included in the revised manuscript. Also, the very relevant aspect of ECG/PPG sensors embedded in wearables and the resulting greater availability of data has been overlooked.
3 - The small changes related to the databases are welcome. However, I completely disagree with the authors when they answer “We leave to the reader to google the individual databases.” The goal of the review is to concentrate and clearly present as much relevant and confirmed information as possible. Leaving important information (in this case, the data available) for the readers to google is regrettable, and there must be relevant details and intricacies on data that the authors (experts) know that are not trivial and would be valuable for current and prospective researchers in AF detection.
4 - This is a very good example of the deep expert insight I would like to see more throughout this manuscript. The readers could find all the technical information on the papers covered by this review, but these portions of overall discussion of the authors’ expert perspective is what makes this review valuable.
Round 3
Reviewer 2 Report
I believe the manuscript is now improved, with deeper discussion and more information that will be useful for readers and prospective researchers in atrial fibrillation monitoring. I thank the authors for answering my comments and adapting the manuscript accordingly.